# A systematic review of Hepatitis B virus (HBV) prevalence and genotypes in Kenya: Data to inform clinical care and health policy

**Louise O. Downs**[1,2], **Cori Campbell**[1], **Paul Yonga**[3], **George Githinji**[4,5], **M. Azim Ansari**[1], **Philippa C. Matthews**[1,6,7,8☯]*, **Anthony O. Etyang**[4☯]

**1** Nuffield Department of Medicine, Medawar Building for Pathogen Research, University of Oxford, Oxford, United Kingdom, **2** Department of Infectious Diseases and Microbiology, John Radcliffe Hospital, Headley Way, Oxford, United Kingdom, **3** CA Medlynks Clinic and Laboratory, Nairobi, and Fountain Projects and Research Office, Fountain Health Care Hospital, Eldoret, Kenya, **4** KEMRI-Wellcome Trust Research Programme, Kilifi, Kenya, **5** Department of Biochemistry and Biotechnology, Pwani University, Kilifi, Kenya, **6** The Francis Crick Institute, London, United Kingdom, **7** Division of Infection and Immunity, University College London, London, London, United Kingdom, **8** Department of Infectious Diseases, University College London Hospital, London, London, United Kingdom

☯ These authors contributed equally to this work.
* philippa.matthews@crick.ac.uk

**Data Availability Statement:** All the data pertinent to the submission are included in the paper and its citations.

## Abstract

The aim of this systematic review and meta-analysis is to evaluate available prevalence and viral sequencing data representing chronic hepatitis B (CHB) infection in Kenya. More than 20% of the global disease burden from CHB is in Africa, however there is minimal high quality seroprevalence data from individual countries and little viral sequencing data available to represent the continent. We undertook a systematic review of the prevalence and genetic data available for hepatitis B virus (HBV) in Kenya using the Preferred Reporting Items for Systematic Review and Meta-analysis (PRISMA) 2020 checklist. We identified 23 studies reporting HBV prevalence and 8 studies that included HBV genetic data published in English between January 2000 and December 2021. We assessed study quality using the Joanna Briggs Institute critical appraisal checklist. Due to study heterogeneity, we divided the studies to represent low, moderate, high and very high-risk for HBV infection, identifying 8, 7, 5 and 3 studies in these groups, respectively. We calculated pooled HBV prevalence within each group and evaluated available sequencing data. Pooled HBV prevalence was 3.4% (95% CI 2.7–4.2%), 6.1% (95% CI 5.1–7.4%), 6.2% (95% CI 4.64–8.2) and 29.2% (95% CI 12.2–55.1), respectively. Study quality was overall low; only three studies detailed sample size calculation and 17/23 studies were cross sectional. Eight studies included genetic information on HBV, with two undertaking whole genome sequencing. Genotype A accounted for 92% of infections. Other genotypes included genotype D (6%), D/E recombinants (1%) or mixed populations (1%). Drug resistance mutations were reported by two studies. There is an urgent need for more high quality seroprevalence and genetic data to represent HBV in Kenya to underpin improved HBV screening, treatment and prevention in order to support progress towards elimination targets.

**Funding:** LD is funded by a Wellcome Clinician PhD fellowship (Grant number BST00070). CC is funded by GlaxoSmithKline (GSK) and the University of Oxford Nuffield Department of Medicine. PCM is funded by Wellcome (ref 110110Z/15/Z), UCL/UCLH NIHR Biomedical Research Centre (BRC) and core funding from the Francis Crick Institute. MAA is supported by a Sir Henry Dale Fellowship jointly funded by the Royal Society and Wellcome (ref 220171/Z/20/Z). For the purpose of open access, the author has applied a CC BY public copyright licence to any Author Accepted Manuscript version arising from this submission. The funders had no role in study design, data collection and analysis, decision to publish, or preparation of the manuscript.

**Competing interests:** I have read the journal's policy and the authors of this manuscript have the following competing interests: CC is partially funded by GlaxoSmithKline. There are no patents, products in development or marketed products associated with this research to declare. This does not alter our adherence to PLOS ONE policies on sharing data and materials.

## Introduction

Chronic hepatitis B (CHB) accounts for an estimated 90,000 deaths annually across West, East and Southern Africa, where most countries are of medium to high prevalence for CHB (prevalence ≥4%), accounting for around 20% of the worldwide burden of infection [1]. The World Health Organisation's (WHO) point prevalence estimate of CHB for Africa is 6.1% (95% CI 4.6–8.5%), but this varies substantially between settings, and high-quality data for individual countries are scarce [1]. CHB meets many of the WHO criteria for a neglected tropical disease, including disproportionately affecting populations living in poverty, being associated with significant stigma and discrimination, and poor investment in clinical infrastructure and research [2]. Fewer than 10% of people have access to testing and treatment, leading to delayed diagnosis, with associated risks of advanced liver disease including hepatocellular carcinoma (HCC) [1].

The Global Health Sector Strategy (GHSS) for viral hepatitis aims to eliminate HBV as a public health threat by 2030 by reducing the incidence of new chronic infections by 90% and reducing mortality by 65% from the 2015 baseline to achieve the 2030 WHO Sustainable Development Goals [3]. These are ambitious targets, and current estimates indicate they will not be attained in most settings until beyond 2050 [4]. Detailed seroprevalence data are lacking, but are urgently needed to target testing, treatment, and prevention interventions to the highest risk groups, to allocate resources, and to inform policy.

In Kenya, there is limited information regarding HBV prevalence. Most studies focus on specific groups such as blood donors and those living with HIV, which may not be representative of the general population [5–7]. Other studies have stringent inclusion criteria, meaning important demographic subgroups remain uncharacterised [8]. HBV testing is not done routinely in Kenya, even in antenatal populations.

Triple HBV vaccine from the age of 6 weeks onwards is recommended by the Kenyan Ministry of Health as a component of the multivalent vaccines rolled out by GAVI within the WHO Expanded Programme for Immunization (EPI). Hep B birth-dose (BD) vaccine for all babies within 24 hours of birth is recommended by the WHO, but has not been adopted by many countries–including Kenya–due to economic and logistical challenges [9]. However, more data are needed to underpin evidence-based policy in this domain, and there is increasing focus on PMTCT as part of 'triple elimination' strategies for HBV/HIV/Syphilis [10].

HBV is divided into 9 genotypes (A-I) with a 10th putative genotype J [11, 12]; these tend to have distinct geographical locations and have been linked to different outcomes. Genotype A predominates in many African countries and has been associated with horizontal transmission, chronicity, early HBeAg seroconversion [13], cirrhosis and HCC development [14]. Genotype also affects response to treatment (including drug resistance), and thus may influence clinical recommendations [13–15], though is not yet widely undertaken in clinical practice in most settings. Most studies of the impact of HBV genotype have been in Asia and Europe. There is a paucity of data on circulating genotypes and subgenotypes in Africa, including Kenya. Whole genome sequencing (WGS) of HBV in Kenya could provide information on transmission networks, disease and treatment outcomes, drug resistance and vaccine escape.

We here assimilate data to describe the seroprevalence and molecular characteristics of HBV infection in Kenya to underpin an evidence-base for local strategies for intervention, and highlight knowledge gaps to inform research. High resolution local data will be essential for development of local clinical care pathways and public health policy, to underpin progress towards the 2030 elimination targets.

## Methods

### Ethics statement

No ethical approval was required for this study.

### Search strategy

We set out to review literature on prevalence and genetic characteristics of HBV infection in Kenya, using the Preferred Reporting Items for Systematic Review and Meta-analysis (PRISMA) 2020 statement checklist (S1 PRISMA Checklist). We searched the online databases PubMed, Embase, African Journals Online (AJOL) and Scopus on 6th December 2021 using the terms in Table 1. We included studies published in English, from 2000 to December 2021 (from 2003 for AJOL) that investigated prevalence, genotype and sequencing of HBV infection in Kenya. We only included data for adults from studies for which the full text was available. There was no minimum number of participants for studies included. We initially screened using a thorough review of the title and abstract, and subsequently reviewed the full manuscripts of eligible articles. Articles that did not meet the inclusion criteria were excluded. Any uncertainty regarding the inclusion of papers was discussed with another reviewer and a consensus obtained.

From each study, we extracted:

- Total number of individuals tested for HBV.

- Number of individuals found to be infected with HBV (either HBsAg positive or HBV DNA positive)

- Study location (city or geographical region)

- Participant selection criteria

**Table 1. Search terms for online databases used during a systematic review on HBV prevalence and genetic information in Kenya.**

| Database searched | Search terms |
|---|---|
| PubMed | (("Hepatitis B"[Mesh]) OR ("hepatitis b"[Title/Abstract] OR HBV[Title/Abstract] OR "hepatitis type b"[Title/Abstract] OR "type b hepatitis"[Title/Abstract] OR "hep b"[Title/Abstract])) AND (("Kenya"[Mesh]) OR (kenya*[Title/Abstract] OR nairobi*[Title/Abstract] OR mombasa*[Title/Abstract])) |
| Embase | 1 exp hepatitis B/ (109266)<br>2 ("hepatitis b" or HBV or "hepatitis type b" or "type b hepatitis" or "hep b").ti,ab. (135330)<br>3 1 or 2 (161216)<br>4 Kenya/ (22999)<br>5 (Kenya* or nairobi* or mombasa*).ti,ab. (26384)<br>6 4 or 5 (29072)<br>7 3 and 6 (227)<br>8 limit 7 to yr = "2000 -Current" (173) |
| AJOL | (("Hepatitis B" OR ("hepatitis b" OR HBV OR "hepatitis type b" OR "type b hepatitis" OR "hep b" AND (("Kenya") OR (Kenya* OR nairobi* OR mombasa*)) |
| Scopus | (("Hepatitis B" OR ("hepatitis b" OR HBV OR "hepatitis type b" OR "type b hepatitis" OR "hep b" AND (("Kenya") OR (Kenya* OR nairobi* OR mombasa*)) |

Terms used to search online databases for a systematic review reporting prevalence and genetic data for HBV in Kenya. We also modified search terms to include the 5 largest cities in Kenya (Nairobi, Mombasa, Eldoret, Nakuru and Kisumu with no change to the results).

- Laboratory methods for confirmation of HBV infection

- Whether any viral sequencing was undertaken, methods used and results (including genotype, presence of vaccine escape and drug resistance mutations).

## Study heterogeneity and HBV risk groups

On the grounds of significant heterogeneity in the populations represented, we divided studies *a priori* into four groups representing populations with differing risks of testing positive for HBV infection. The low-risk group included studies likely to be most representative of the general population (antenatal women, healthcare workers, blood donors and the national survey). The moderate risk group consisted of studies containing populations living with HIV. High-risk groups were defined as people with risk factors for acquisition of blood-borne virus infection, including people who inject drugs, men who have sex with men (MSM) and sex workers. Those presenting to hospital with hepatitis or jaundice were defined as a very high-risk group, as HBV infection is enriched in populations presenting with established liver disease, particularly if the background population has medium or high HBV prevalence. This risk stratification system is a pragmatic approach to a highly heterogenous literature and we have used these risk groups for ease of reference throughout this review.

## Quality assessment of studies

A thorough assessment of the study quality was done using the PRISMA guidelines [16] and Joanna Briggs Institute critical appraisal checklist for prevalence studies (S1 Table) [17]. Any dispute surrounding study quality was discussed with another reviewer and a consensus reached.

## Identifying and analysing full-length HBV sequences from Kenya

We downloaded all full genome HBV sequences from Kenya in GenBank on 1-December-2021 to assimilate a reference set of all whole genome sequences representing Kenya. Sequences were aligned with available HBV reference sequences for each genotype (11) using MAFFT [18]. A maximum likelihood phylogenetic tree with bootstrap replicates of 1000 was created using NGPhylogeny.fr [19].

## Statistical analysis

Pooled prevalence estimates for CHB infection (i.e. HBsAg seroprevalence) were generated via meta-analysis using logit transformed proportions of standard errors. Meta-analysis was conducted in R (version 4.1.1) using the 'meta' package (version 5.2–1). Standard errors for prevalence were calculated using the formula $\sqrt{p(1-p)/n}$, with the total sample size represented by $n$ and the proportion of the sample seropositive for HBV represented by $p$. Meta-analysis was stratified by cohort risk group. The $I^2$ statistic was used to quantify the heterogeneity between studies within each risk group, calculated using $I^2 = 100\% \times (Q\text{-df})/Q$ where Q–Cochran's coefficient and df = degrees of freedom. Pooled estimates from the random effects models are presented for each risk group. Study site locations were identified using latitudinal and longitudinal data from Googlemaps based on where the study was conducted. The map plot was created in R (version 4.1.1) using 'rworldmap' (version 1.3–6) 'ggmap' (version 3) and 'ggplot2' (version 3.3.5) packages.

### Occult HBV infection

Occult HBV infection (OBI) is defined as detectable HBV DNA in the absence of HBsAg. Where studies reported both HBsAg positivity rates and OBI rates in those who were HBsAg negative, only prevalence data based on HBsAg positivity was included in the meta-analysis, in order to ensure datasets were comparable between studies.

## Results

### (i) Identification of studies

We identified 272 published studies, of which 23 studies met the inclusion criteria for prevalence assessment, representing a total of 11,467 people (Fig 1 and Table 2). Three of these studies also screened individuals for occult HBV infection (OBI) in a total of 666 people using HBV DNA polymerase chain reaction (PCR) in addition to testing for HBsAg seroprevalence. Two studies screened initially with HBsAg, then with HBV DNA PCR on those who were HBsAg negative [20, 21]. A further study included two different populations: a) those attending a clinic for sex workers, whom they screened initially for HBsAg, then HBV PCR in those who were HBsAg negative and b) known HBsAg negative, jaundiced participants whom they screened with HBV DNA PCR to detect OBI [22].

We identified nine studies reporting HBV sequence data (full or partial genome), including seven studies from among the 23 seroprevalence studies described above (Table 2), and an additional two studies that only included HBsAg positive participants so were not included in prevalence analysis [23, 24]. One study did not clearly report how many HBV samples were sequenced or the genotyping results, and this study was excluded from further analysis [25]. Eight studies remained for analysis representing 247 individuals (Table 2).

We identified eight studies reporting HBsAg prevalence in low-risk populations (total number of individuals = 6828), seven studies in people living with HIV (medium risk, total number of individuals = 1861), five studies in high-risk groups (total number of individuals = 2221) and three studies in people presenting to clinical services with established liver disease (defined here as very high-risk for HBV infection; total number of individuals = 492).

### (ii) Geographical distribution of HBV seroprevalence data

Of the 23 studies included, 14 (61%) were in Nairobi or Mombasa, Kenya's most populous cities (Table 2), and all studies were done in the South of the country along the infrastructure routes between Mombasa, Nairobi and Kisumu. These are also the most densely populated Kenyan counties [48]. Kisumu was the city most represented in the studies by overall sample size (Fig 2).

The mean cohort sample size was 599 participants (IQR 434). 14 studies recruited participants for cohort inclusion at outpatient clinics (8 in HIV clinics, 4 in blood donor clinics, 1 in a health clinic and 1 in antenatal clinic), one captured data through the blood donor registry, three undertook community outreach screening, three recruited hospital inpatients, one recruited healthcare workers and one was a national survey of urban and rural population groups (Table 2).

### (iii) Quality assessment of the literature

Overall the quality of studies investigating HBV prevalence in Kenya was low (Fig 3 and S1 Table). 17/23 studies were cross sectional, reporting HBV population prevalence at a single time point only. Most cohort sampling methods were non-randomised and only 4/21 studies detailed their sample size calculation [20, 21, 37, 41]. Several studies sampled people only from

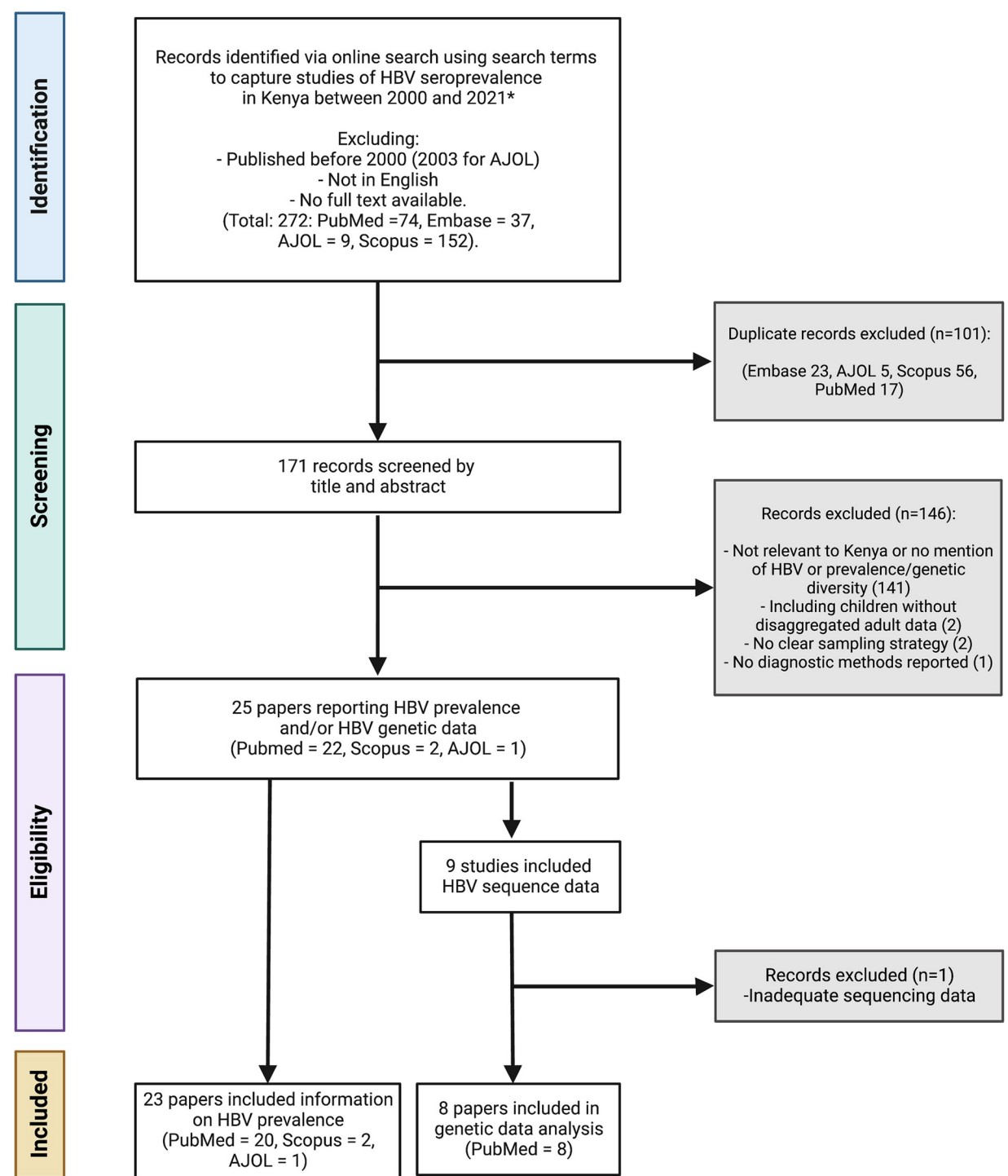

**Fig 1. Flow chart for eligibility of studies included in a systematic review reporting prevalence and genetic data for HBV in Kenya between 2000–2021.** (AJOL: African Journal Online). All eight studies included for genetic analysis contain information on HBV genotype. Figure created in Biorender.com with licence to publish.

**Table 2. Characteristics of the studies included in the systematic review reporting prevalence and genetic data for HBV in Kenya.**

| First Author | Year | N | Location | HBV +ve (%) | Data collection method | Study design | Sampling type | Population recruitment location | Population Investigated | Detection Method |
|---|---|---|---|---|---|---|---|---|---|---|
| **Very high-risk populations (total 3 studies representing 492 individuals)** | | | | | | | | | | |
| Atina JO [26] | 2004 | 60 | Nairobi | 30 | Prospective | Cross sectional | Non-random | Hospital | Hepatitis | RPHA |
| Muchiri I [27] | 2013 | 100 | Nairobi | 13 | Prospective | Cross sectional | Non-random | Hospital | Hepatitis | RPHA |
| Ochwoto M* [28] | 2016 | 332 | Nairobi, Mombasa, Kisumu Eldoret | 50.6 | Prospective | Cross sectional | Non-random | Hospital | Jaundiced | CLEIA |
| **High risk populations (total 5 studies representing 2221 individuals)** | | | | | | | | | | |
| Day SL [7] | 2013 | 159 | Mombasa | 7 | Prospective | Cohort | Non-random | HIV clinic | Female sex workers | CLEIA |
| Webale MK* [29] | 2015 | 752 | Mombasa | 5.2 | Retrospective | Cross sectional | Non-random | Community | Drug users | 5 panel rapid test |
| Wahome E [30] | 2016 | 525 | Coastal Kenya | 8 | Retrospective | Cohort | Non-random | Community | HIV neg MSM | ELISA |
| Oyaro M* [31] | 2018 | 673 | Nairobi, Mombasa, Kisumu | 4.3 | Prospective | Cross sectional | Non-random | Community | Drug users | ELISA |
| Jepkemei [22] | 2020 | 112 | Nairobi | 9.8 | Retrospective | Cohort | Non-random | Health clinic | MSM | CLEIA |
| **Moderate risk populations (total 7 studies representing 1861 individuals)** | | | | | | | | | | |
| Harania RS [32] | 2008 | 357 | Nairobi | 6.2 | Prospective | Cross sectional | Non-random | HIV clinic | Adults HIV | ELISA |
| Kim HN* [33] | 2011 | 389 | Nairobi | 6.9 | Retrospective | RCT | Random | HIV clinic | Adults HIV | ELISA |
| Muriuki BM [34] | 2013 | 290 | Nairobi | 6.2 | Prospective | Cross sectional | Non-random | HIV clinic | Adults HIV | ELISA |
| EN Njuguna [35] | 2015 | 322 | Nairobi, Thika | 1 | Prospective | Cross sectional | Random | HIV clinic | HIV discordant couples | ELISA |
| Maina DN [36] | 2017 | 190 | Kajiado | 5.8 | Prospective | Cross sectional | Non-random | HIV clinic | Adults HIV | CLEIA |
| Greer AE [25] | 2017 | 105 | Kisumu | 11 | Prospective | RCT | Non-random | HIV clinic | HIV discordant couples | CLEIA |
| Salyani A [21] | 2021 | 208 | Nairobi, Kijabe, Mbagathi | 5.8 | Prospective | Cross sectional | Non-random | HIV clinic | Adults HIV | CLEIA |
| **Low risk populations (total 8 studies representing 6828 individuals)** | | | | | | | | | | |
| Ngaira JA [37] | 2016 | 287 | Mbagathi | 3.8 | Prospective | Cross sectional | Random | Antenatal clinic | Pregnant Women | ELISA |
| Ly KN [38] | 2016 | 1091 | Countrywide | 2.1 | Retrospective | 2 stage cluster | Random | National survey | Urban and rural clusters | ELISA |
| Wamamba D [39] | 2017 | 2046 | Kisumu | 3.1 | Retrospective | Cross sectional | Non-random | Blood donor clinic | Blood Donors | ELISA |
| Onyango CG [40] | 2018 | 1215 | Kisumu, Siaya, Homa Bay | 3.5 | Prospective | Cross sectional | Random | Blood donor clinic | Blood Donors | ELISA |
| Bartonjo G [5] | 2019 | 594 | Nakuru, Tenwek | 5.6 | Prospective | Cross sectional | Random | Blood donor registry | Blood Donors | ELISA |
| Kisangau EN [41] | 2019 | 295 | Makueni | 4.5 | Prospective | Cross sectional | Random | Hospital rota | Healthcare workers | ELISA |
| Awili HO [42] | 2020 | 1000 | Kisumu, Siaya, Homa Bay | 3.4 | Prospective | Cross sectional | Non-random | Blood donor clinic | Blood Donors | ELISA |
| Aluora PO [20] | 2020 | 300 | Nairobi | 2.3 | Prospective | Cross sectional | Random | Blood donor clinic | Blood Donors | CLEIA |
| **Studies of known HBsAg positive people included for genetic data analysis (total 2 studies representing 90 individuals)** | | | | | | | | | | |

*(Continued)*

**Table 2.** (Continued)

| First Author | Year | N | Location | HBV +ve (%) | Data collection method | Study design | Sampling type | Population recruitment location | Population Investigated | Detection Method |
|---|---|---|---|---|---|---|---|---|---|---|
| Kwange SO* [23] | 2013 | 32 | Nairobi | 100 | Retrospective | Cross sectional | | Blood donor registry | Blood Donors | ELISA |
| Ochwoto* [24] | 2013 | 58 | Nairobi | 100 | Prospective | Cross sectional | | Blood donor registry | Blood Donors | ELISA |
| **Studies of occult HBV infection (OBI) prevalence in HBsAg negative individuals (total 3 studies representing 666 individuals)** | | | | | | | | | | |
| Jepkemei* [22] | 2020 | 166 | Nairobi | 18.7 | Retrospective on HBsAg -ve only | Cohort | Non-random | Health clinic | MSM/Jaundice | HBV DNA PCR |
| Salyani A [21] | 2021 | 196 | Nairobi, Kijabe, Mbagathi | 5.3 | Retrospective on HBsAg -ve only | Cross sectional | Non-random | HIV clinic | Adults HIV | HBV DNA PCR |
| Aluora PO* [20] | 2020 | 293 | Nairobi | 2.4 | Retrospective on HBsAg -ve only | Cross sectional | Random | Blood donor clinic | Blood Donors | HBV DNA PCR |

This includes two studies on solely HBsAg +ve patients used only for analysis of genetic data [23, 24] and three studies of Occult HBV infection (OBI) prevalence [20–22]. Studies are ranked as very high / high / moderate / low risk groups (presented in order) and within these categories ordered by date of publication.

Studies marked with an asterisk (*) are those which presented HBV genetic information including genotype.

N–number in the study; RCT–randomised controlled trial; RPHA–Reverse passive haemagglutination; CLEIA–Chemiluminescent immunoassay; ELISA–enzyme linked immunosorbent assay; PCR–polymerase chain reaction; HBsAg–Hepatitis B surface antigen; MSM–men who have sex with men.

small geographical locations or from a subset of the general population e.g. HIV negative individuals. 21/23 studies used either an enzyme linked immunosorbent assay or chemiluminescent enzyme immunoassay (ELISA or CLEIA) for HBsAg diagnosis. Two studies used reverse passive haemagglutination for diagnosis of CHB, a method previously demonstrated to have poor sensitivity [26, 27, 43] (Table 2). 2/23 studies went on to screen the HBsAg negative population for HBV DNA via PCR [20, 21] and one study included a known HBsAg negative population which they screened for HBV DNA [22].

## (v) HBV prevalence estimates in different risk groups

The pooled estimate for HBV prevalence using a random effects model in the low-risk group was 3.36% (95% CI 2.67–4.21%) compared with 6.14% in the moderate risk group (95% CI 5.08–7.41%), 6.18% (95% CI 4.6–8.19%) in the high-risk group and 29.19% (95% CI 12.15–55.14%) in the very high-risk group, however we note that the confidence interval of this estimate is very wide (Fig 4). Heterogeneity was significant ($I^2 > 50\%$) within each subgroup, and highest in the very high-risk sub-group ($I^2 = 95\%$, $p < 0.01$).

Three studies screened for OBI using HBV DNA PCR. These were in populations known to be HBsAg negative and from different HBV risk groups: blood donors, those living with HIV and those presenting to hospital with jaundice. OBI prevalence estimates in these studies were 2.4%, 5.3% and 18.7% respectively [20–22].

## (vi) Identification of HBV sequences

All eight studies including HBV genetic information used PCR of the HBV basal core promotor, Pol or S genes for amplification, followed by Sanger sequencing to determine genotype. Two studies looked for known drug resistance-associated mutations (RAMs) [23, 24]. Two studies undertook whole genome HBV sequencing in a total of 22 patients [23, 28]. 228/247 (92%) of participants were infected with HBV genotype A, 15/247 (6%) with genotype D infection, whilst the remaining were either mixed genotype populations (2/247) or genotype D/E

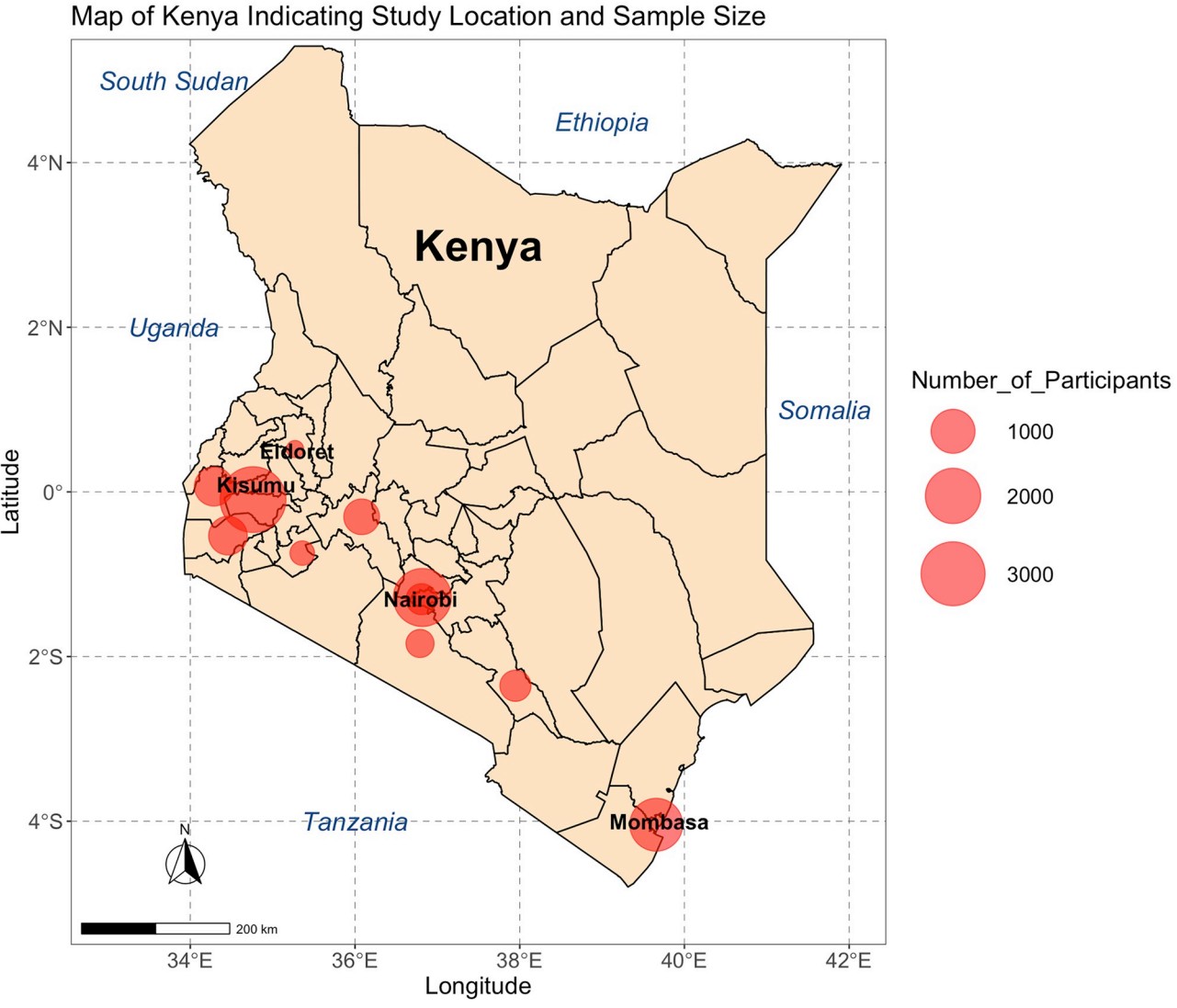

**Fig 2. Map of Kenya indicating the locations and size of populations from which seroprevalence of HBV infection is reported.** Data from a systematic review of papers reporting prevalence and genetic data for HBV in Kenya between 2000 and 2021. The size of the red circle indicates numbers screened in each location, studies in the same location are grouped together. n = number of individuals reported. Surrounding countries are marked in blue, Kenya's four most populous cities are marked in black. Figure created using R version 4.2.0, packages ggmaps version 3.0.0, ggplot2 version 3.3.6 and sf version 1.0–7. The Kenyan county shapefiles were obtained from the Humanitarian Data Exchange, available open source from https://data.humdata.org/dataset/geoboundaries-admin-boundaries-for-kenya.

recombinants (2/247) (Table 3). Sub-genotype was determined in 146/247 (59%) participants. This was most commonly sub-genotype A1 (134/146, 92%) in keeping with previous regional data [44].

To provide further background context for HBV sequences in Kenya, we identified 25 full length HBV sequences from GenBank (Fig 5). These were generated from three studies, published in 2013, 2015 and 2016 [24, 28, 45]. They primarily represented individuals presenting to hospital with jaundice (21/25 sequences), infected with genotypes A1 and D.

5/8 studies provided a detailed analysis of either amino acid or nucleotide substitutions found in the sequenced region of HBV [20, 23, 24, 33, 37]. 2/5 studies correlated these with

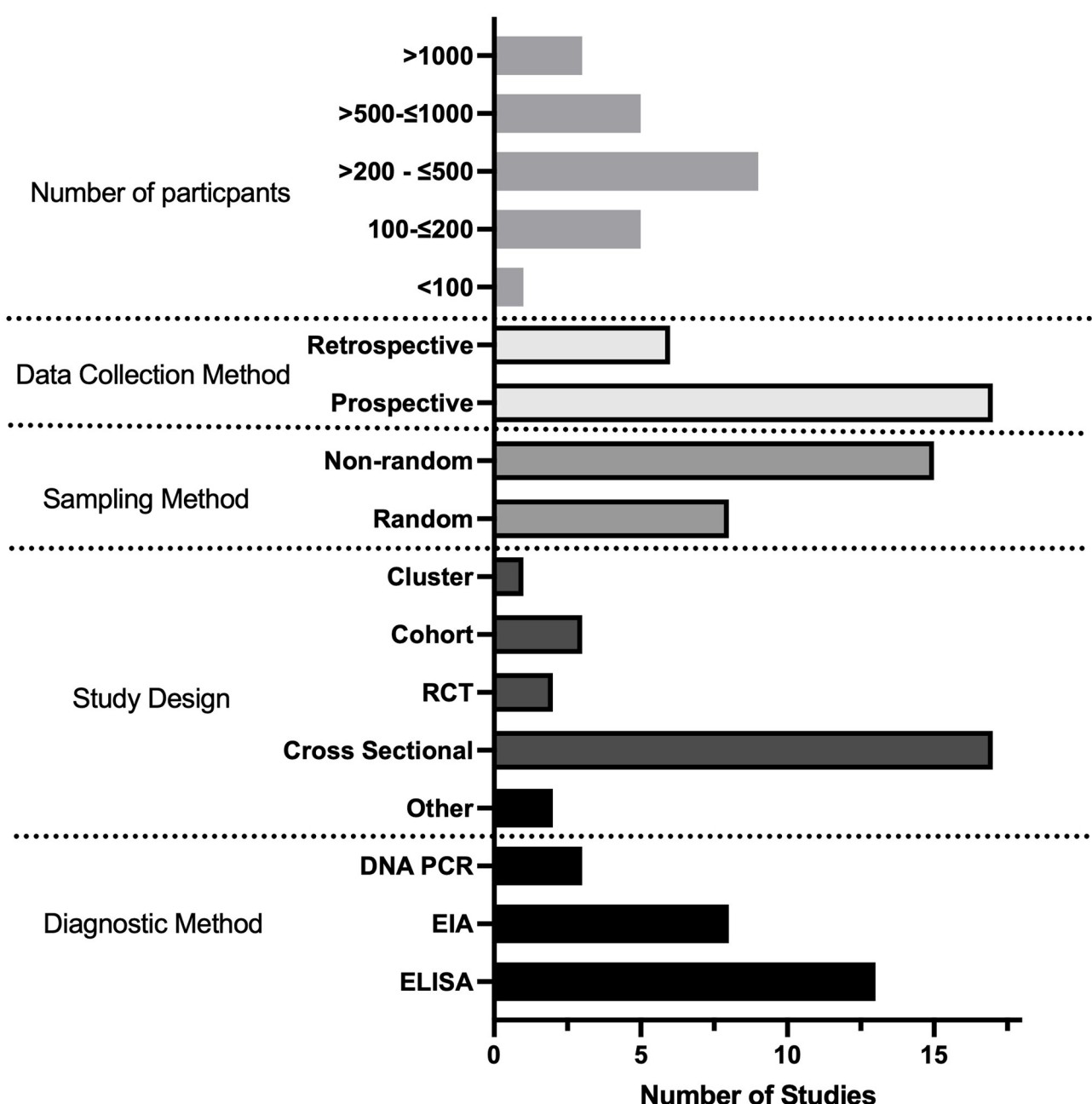

**Fig 3. Bar chart showing characteristics of studies identified for a systematic review reporting prevalence and genetic data for HBV in Kenya.**
This is stratified by number of participants, study design, sampling method, data collection and diagnostic methods. RCT: Randomised controlled trial; EIA: Chemiluminescent enzyme immunoassay; ELISA: Enzyme linked immunosorbent assay.

known drug resistance mutations to lamivudine and other nucleoside analogues (Table 4) [20, 33]. One study reported the emergence of drug resistance mutations during lamivudine treatment associated with breakthrough HBV viraemia [33]. Multiple other mutations were described in the five studies, some of which were in the major hydrophilic region of the surface gene, and thus potentially important in influencing both natural and vaccine-mediated immunity [46, 47].

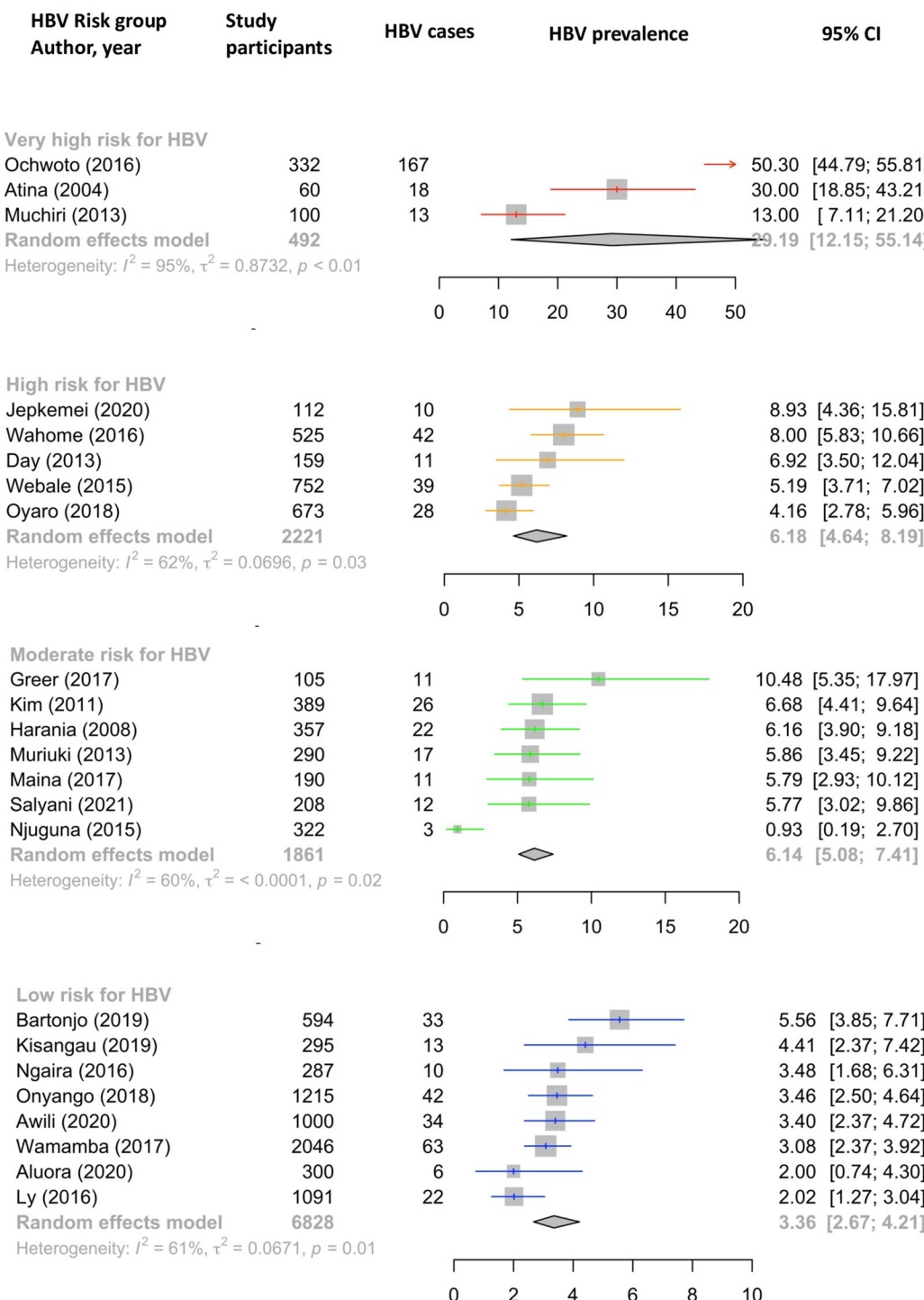

**Fig 4. Forest plot showing pooled HBV prevalence for populations in different risk groups by random effects model.** Data generated through a systematic review reporting prevalence and genetic data for HBV in Kenya between 2000–2021. In each case, the size of the population included is represented by the size of the square. Point prevalence and 95% Confidence Interval (CI) is indicated for each study. Studies are ordered by HBV prevalence in each risk group.

**Table 3. Numbers of individuals in which HBV genotype or sub-genotype was reported for study populations in Kenya, as part of a systematic review for HBV in Kenya from 2000–2021.** Data from 8 studies marked * in Table 2.

| Genotype | Sub-genotype | Number of individuals |
|---|---|---|
| A (total 228) | Not sub-genotyped | 94 |
| | A1 | 134 |
| D (total 15) | Not sub-genotyped | 3 |
| | D3 | 3 |
| | D4 or D6 | 9 |
| D/E Recombinants | | 2 |
| Unclear | | 2 |

## (vi) HBV serology and HBV biomarkers

Exploring the prevalence of anti-HBs (vaccination or exposure) and anti-HBc (exposure to HBV) in HBsAg-negative populations is important to build up a full picture of population epidemiology. Among individuals testing HBsAg-positive, a panel of biomarkers is used to determine treatment eligibility, including HBeAg status, HBV DNA viral load, liver enzymes and imaging scores. These parameters are outside the primary scope of this study, but the data can be accessed as a supporting data file [48].

## Discussion

Enhanced efforts to characterise the epidemiology and disease burden of HBV are urgently required in Africa, as HBV is present at medium to high endemicity in many populations but has been neglected as a public health problem. Here we have reviewed the literature available on prevalence, genotypes and drug resistance data for CHB in Kenya. In our 'low-risk' category, intended to provide estimates most reflective of the general population, the pooled prevalence estimate for HBV infection was 3.4%. Point-prevalence estimates of ~6% were obtained for the groups we defined as medium and high risk, comprising people living with HIV infection and those with other identified risk factors for blood-borne virus infection. Similar prevalence estimates in the moderate- and high-risk groups was only evident after analysis. The number of studies was too low to allow for further subdivision into individual risk groups (e.g. comparing people who inject drugs, MSM, and sex workers). In the population presenting to healthcare facilities with established symptomatic liver disease (classified here as 'very high risk'), the prevalence of HBV was 29.2% (although the underlying primary risk factor(s) for HBV acquisition in this group are not established).

In this very high-risk group, wide confidence intervals along with significant heterogeneity ($I^2 = 95\%$) are notable. This population evidently has very different pre-test probabilities for HBV infection depending on underlying risk factors. In the absence of robust screening programmes, many people do not find out they have HBV infection until presenting to hospital with manifestations of liver disease. While the prevalence in this group evidently cannot be extrapolated to the general population, it is nevertheless an important observation that HBV in this setting accounts for such a high proportion of end-stage liver disease. Furthermore 2/3 studies in this very high-risk group used RPHA for HBsAg detection which is less sensitive than HBsAg, and therefore may underestimate true prevalence of HBV infection.

Most studies included in this review focussed on specific groups of people such as blood donors and those co-infected with HIV. Blood donation in Kenya is voluntary and often done by family members of those in need. There is no financial compensation for donation [49]. Routine screening for HBV through the Kenyan National Blood Transfusion Service (KNBTS)

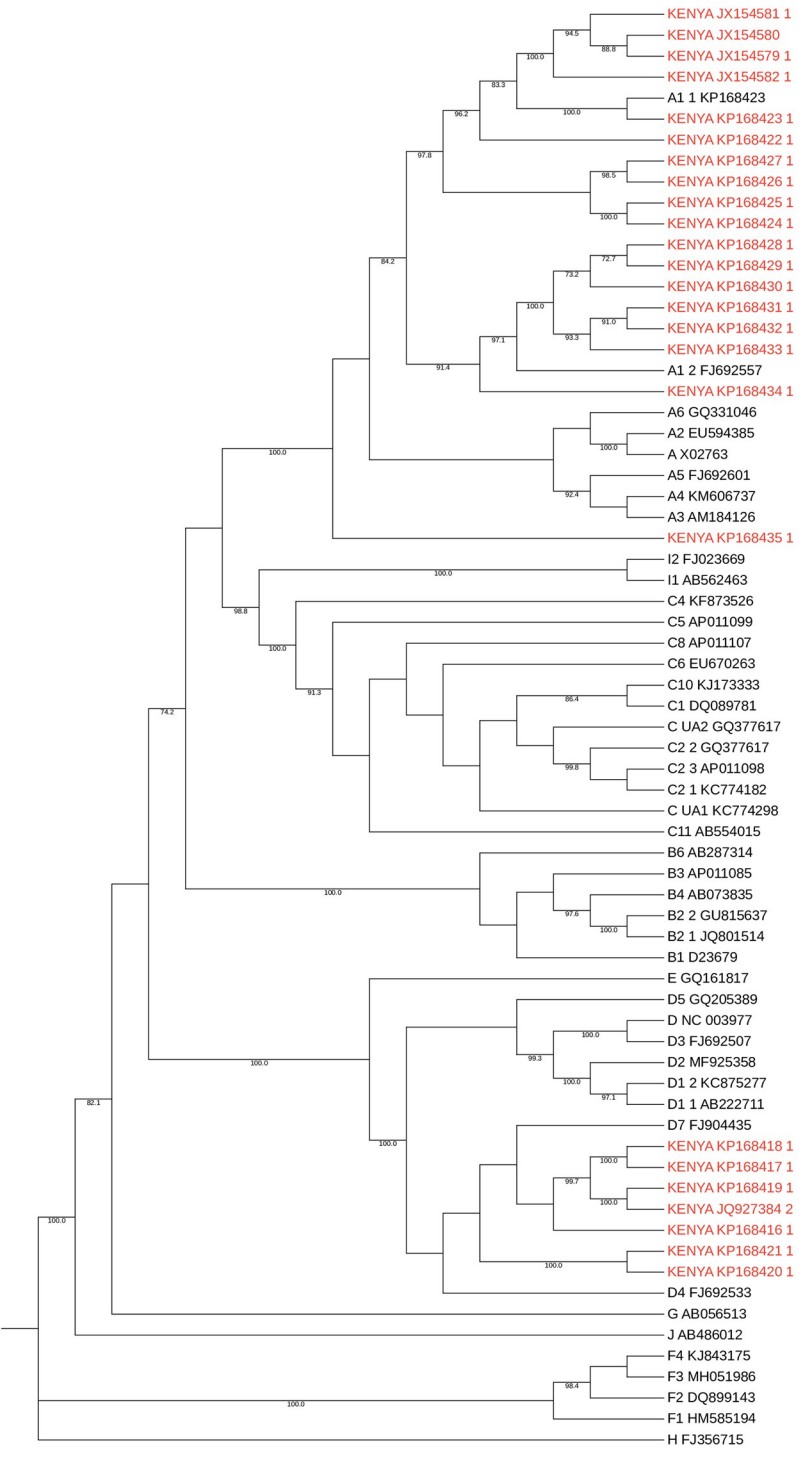

**Fig 5. Maximum likelihood phylogenetic tree of full-length consensus HBV sequences from Kenya.** Kenyan sequences are those published in GenBank (downloaded 1st Dec 2021) and are shown in red alongside genotype reference sequences in black (1000 bootstrap replicates were performed, and bootstrap support of ≥70% are indicated. Reference sequences from McNaughton et al. (2020) [11].

**Table 4. Details of HBV mutations found in two studies in Kenya which reported presence of drug resistance mutations, as part of a systematic review on prevalence and genetic data for HBV in Kenya from 2000–2021.**

| First Author | Publication Year and number of participants | Mutations Identified | Number of individuals with mutation | Resistance conferred |
|---|---|---|---|---|
| Kim HN [33] | 2011 (n = 389) 21 participants had HBV sequenced | rt1233V rtA181S rtV207I | 3/21 had rt1233V, rtA181S or rtV207I prior to treatment | Adefovir + lamivudine |
| | | A200V M204I | 1/21 developed A200V + M204I during lamivudine treatment | Lamivudine |
| | | M204I | 1/21 developed M204I during lamivudine treatment | Lamivudine |
| Patrick Okoti Aluora [20] | 2020 (n = 300) 9 participants had HBV sequenced (2 HBsAg +ve, 7 HBV DNA +ve only) | rt181T | 1/9 | Lamivudine, Adefovir |
| | | M129L V163I | 9/9 | Lamivudine |

In Kim HN, two patients developed HBV breakthrough viraemia during lamivudine treatment and developed either RT M204I mutation alone or both A200V and M204I mutations during treatment.

consists of ELISA for HBsAg only and there is no nucleic acid amplification testing (NAAT); some OBI may therefore go unidentified. Only one study in this review focussed on pregnant women [37] and one study enrolled healthcare workers [41]. These are accessible and important groups to screen for HBV infection given they are engaged with healthcare, likely to come for follow up visits, and interventions can have a significant impact on reducing transmission events. Treatment for pregnant mothers and healthcare workers would reduce onward transmission, and vaccination uninfected healthcare workers and babies at birth would decrease the overall burden of infection, reducing morbidity and mortality. One study was nationwide [38], but only included those who were HIV negative. More general population screening is lacking, and testing is not routinely done when presenting to healthcare facilities [50]. Some areas of Kenya have been more rigorous in their diagnostic approaches, but this is sporadic and may be increased only when there is a known outbreak of HBV in the local community, as has been the case in other African countries [51, 52]. This may give a skewed view on population prevalence, but also leads to missed opportunities for diagnosis and intervention, particularly given the very high proportion of those presenting to hospital with jaundice or hepatitis found to be infected with HBV (pooled HBV prevalence 29.19% and 18.7% OBI prevalence).

It is notable that no studies were done in Northern Kenya, particularly along the borders with Somalia and South Sudan where the prevalence of HBV is likely to be substantially higher (for these two neighbouring countries, HBsAg prevalence is estimated at 19% and 12% respectively [53, 54], however population density here is also very low [55].

Along with minimal population screening, there is very little sequencing of HBV in Kenya. Among the 25 papers we reviewed regarding HBV sequencing, only two reported whole genome sequencing, and none did next generation sequencing. We identified only 25 complete HBV genomes from Kenya in a GenBank search. Most available data is from single gene PCR and Sanger sequencing of S and P genes to determine genotype. Expanding these data will allow identification of recombinant genotypes, of which there is evidence in Kenya [28, 31], but currently without good understanding of how these translate into clinical outcomes. Deep sequencing data will enable detection of minority variant mutations that may be relevant in emergence of vaccine escape and drug resistance, and also allow description of viral quasispecies, how this correlates with clinical phenotype and other biomarkers.

Three studies reviewed here screened for OBI using PCR. OBI prevalence was similar to estimated pooled HBsAg prevalence in the associated risk group (2.4%, 5.3% and 18.7% OBI

prevalence in low, medium and high/very-risk groups compared with 3.36%, 6.14% and 6.18/29.19% pooled HBsAg positivity estimates in the equivalent groups). This indicates that many HBV cases are being missed due to the lack of appropriate screening tests, however the cost and poor availability of HBV DNA testing means it is not currently feasible to use as a universal screening test in Kenya. 20/23 studies solely reported HBsAg positivity diagnosed using other less sensitive tests. It is worth noting that of those presenting to hospital with jaundice who were HBsAg negative, nearly 20% were HBV DNA positive. It is not known whether the jaundice was due to acute HBV infection, or reactivation of chronic disease, but it seems to be an important indicator of HBV infection and screening of all those presenting to hospital with jaundice or hepatitis for OBI with HBV DNA PCR would be optimal. Few studies had characterised HBV exposure and vaccination status using anti-HBc and anti-HBs respectively. This highlights a broader issue around funding and access to laboratory tests needed for complete epidemiological assessment of populations.

## HIV coinfection as a special case

The prevalence of HIV infection in adults in Kenya is 4.2% (95% CI 3.7–4.9%) [56]. Seven studies included in this analysis reported HBV prevalence in people living with HIV. The pooled HBV prevalence in this group was 6.14% (95% CI 5.08–7.41%). The HIV population is better represented than other groups at risk, as HBV screening is easier to offer to individuals already accessing healthcare for HIV monitoring and treatment. Through this established infrastructure for HIV (including clinics with staff, laboratory support, blood monitoring and drug distribution services), clinical care pathways for HBV could be incorporated. Although tenofovir is available free of charge in Kenya and is on the WHO list of essential medicines [57], it is only consistently available in combination with lamivudine or emtricitabine for HIV treatment, leaving the HBV monoinfected population unable to access licensed monotherapy.

## Limitations

The HBV prevalence estimates we have generated here are wide and vary significantly between the risk groups (pooled risk group prevalence 3.36% - 29.19%). The very high-risk group also has a very wide confidence interval for prevalence estimates. Our risk groups were determined *a priori* based on existing understanding of the distribution of HBV infection, but data were insufficient to disaggregate into more specific groups, and we recognise that the prevalence of HBV infection in populations at risk varies substantially by setting. Other sources have different estimates of Kenyan HBV prevalence (e.g. 1% by the CDA Foundation [4]). The CDA data are from 2016, so may be out of date, but the varying estimates reflect difficulties with methods of data collection, varying data sources and data missingness. The overall quality of studies was low, with non-random sample selection common, no calculation of sample size in most studies and nearly all studies being cross sectional representing only a snapshot of HBV prevalence. Only selected populations are represented by the studies we identified, and even those studies seeming to represent the population more broadly are subject to bias. For example, the study of healthcare workers was primarily female nurses [41] and the nationwide survey only included HIV negative participants [38]. We considered only including those studies reaching a certain quality threshold in the prevalence meta-analysis, however this would have substantially restricted the available data. For example, including only those studies with random sampling methods and a documented sample size calculation would have left only three studies. One of the key findings of this systematic review is the lack of good quality seroprevalence data, and detailing this gives a good understanding of available literature.

There are no data for the northern part of Kenya, including the region around the border with South Sudan where there might be migration of high prevalence populations. It is likely that prevalence of HBV infection varies significantly by age, region of the country, and according to particular at-risk groups–thus targeted surveillance is important to provide an evidence-base for local and population-specific interventions.

No children were included in this review. In 2019 Kenya achieved an average coverage of 91% of 3$^{rd}$ dose HBV childhood vaccination [58], but in future studies, screening children for HBsAg, anti-HBc and anti-HBs by birth cohort would be important to determine the impact of the vaccine campaign on infection, exposure and immunity, and to identify any populations being missed by vaccine coverage. There are increasing calls for the scale-up of BD HBV immunisation as part of a triple elimination campaign.

We highlight the poor representation of HBV in Kenya with sequencing data, identifying only two studies that undertook whole genome sequencing. 24/25 sequences available on Gen-Bank were from two studies. This is clearly not representative of HBV in the general population, and work is required to determine circulating genotypes and to characterise polymorphisms that are relevant to outcomes of infection, treatment and vaccination.

## Conclusions

We have assimilated epidemiological data for HBV in Kenya, together with genetic parameters where available, to provide the most refined picture possible to date. Our data suggest that Kenya falls into the 'intermediate' prevalence group (2–5%, as defined by the WHO). A sparse literature highlights the pressing need for clinical and research enterprise, to provide an evidence base for realistic and practical strategies that support country-specific scale-up of screening and treatment. Alongside continued efforts for three-dose vaccine coverage in infancy, enhanced interventions may include focus on HBV birth dose vaccine as part of the triple elimination initiative, with improved access to diagnostics, surveillance and treatment, to curtail the burden of disease in those currently infected, and reduce the incidence of new infections, moving Kenya towards 2030 elimination targets.

## Supporting information

**S1 Checklist. Preferred Reporting Items for Systematic Review and Meta-analysis (PRISMA) 2020 statement checklist.**
(DOCX)

**S1 Table. Joanna Briggs critical appraisal checklist.**
(DOCX)

## Acknowledgments

This manuscript was written with the permission of the Director, KEMRI-CGMRC.

## Author Contributions

**Conceptualization:** Louise O. Downs, Philippa C. Matthews, Anthony O. Etyang.

**Data curation:** Louise O. Downs.

**Formal analysis:** Louise O. Downs, Cori Campbell, M. Azim Ansari.

**Funding acquisition:** Louise O. Downs, Philippa C. Matthews.

**Investigation:** Louise O. Downs, Paul Yonga, George Githinji, Philippa C. Matthews, Anthony O. Etyang.

**Methodology:** Louise O. Downs, Cori Campbell, M. Azim Ansari, Philippa C. Matthews.

**Project administration:** Louise O. Downs.

**Supervision:** M. Azim Ansari, Philippa C. Matthews, Anthony O. Etyang.

**Writing – original draft:** Louise O. Downs.

**Writing – review & editing:** Cori Campbell, Paul Yonga, George Githinji, M. Azim Ansari, Philippa C. Matthews, Anthony O. Etyang.

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
