## [Decision Letter · Decision Letter 0]

26 Aug 2022

PGPH-D-22-00830

A systematic review of Hepatitis B virus (HBV) prevalence and genotypes in Kenya: Data to inform clinical care and health policy

Dear Dr. Matthews,

Thank you for submitting your manuscript to PLOS Global Public Health. After careful consideration, we feel that it has merit but does not fully meet PLOS Global Public Health’s publication criteria as it currently stands. Therefore, we invite you to submit a revised version of the manuscript that addresses the points raised during the review process.

We look forward to receiving your revised manuscript.

Kind regards,

Abraham D. Flaxman, Ph.D.

Academic Editor

Journal Requirements:

1. Please send a completed 'Competing Interests' statement, including any COIs declared by your co-authors. If you have no competing interests to declare, please state "The authors have declared that no competing interests exist". Otherwise please declare all competing interests beginning with the statement "I have read the journal's policy and the authors of this manuscript have the following competing interests:"

2. Please insert an Ethics Statement at the beginning of your Methods section, under a subheading 'Ethics Statement'. It must include:

  a. The name(s) of the Institutional Review Board(s) or Ethics Committee(s)

  b. The approval number(s), or a statement that approval was granted by the named board(s) 

  c. (for human participants/donors) - A statement that formal consent was obtained (must state whether verbal/written) OR the reason consent was not obtained (e.g. anonymity). NOTE: If child participants, the statement must declare that formal consent was obtained from the parent/guardian.]

3. We ask that a manuscript source file is provided at Revision. Please upload your manuscript file as a .doc, .docx, .rtf or .tex.

4. Figure 2: please (a) provide a direct link to the base layer of the map (i.e., the country or region border shape) and ensure this is also included in the figure legend; and (b) provide a link to the terms of use / license information for the base layer image or shapefile. We cannot publish proprietary or copyrighted maps (e.g. Google Maps, Mapquest) and the terms of use for your map base layer must be compatible with our CC-BY 4.0 license. 

Additional Editor Comments (if provided):

Reviewers' comments:

Reviewer's Responses to Questions

**Comments to the Author**

1. Does this manuscript meet PLOS Global Public Health’s publication criteria? Is the manuscript technically sound, and do the data support the conclusions? The manuscript must describe methodologically and ethically rigorous research with conclusions that are appropriately drawn based on the data presented.

Reviewer #1: Yes

Reviewer #2: Partly

2. Has the statistical analysis been performed appropriately and rigorously?

Reviewer #1: Yes

Reviewer #2: Yes

3. Have the authors made all data underlying the findings in their manuscript fully available (please refer to the Data Availability Statement at the start of the manuscript PDF file)?

Reviewer #1: Yes

Reviewer #2: Yes

4. Is the manuscript presented in an intelligible fashion and written in standard English?

Reviewer #1: Yes

Reviewer #2: Yes

5. Review Comments to the Author

Reviewer #1: This is a much needed review that has great potential to generate hypothesis and influence policy. Diagnosing Hepatitis B is a rather error free process and "elimination of HBV by 2030" will only be made possible by testing more people and vaccination. Quality of evidence notwithstanding, the information discussed in this paper is of great value.

Abstract

The authors report that studies reviewed are low quality. these studies are cross sectional which in its very nature is a limiting study design. Is it possible for the abstract to have subtitles? It not clear what the objective of the study is in the abstract. The result section on the abstract is too long. the recommendation/conclusion not congruent to the study findings.. make recommendations based on the study findings of prevalence of 3% to 29% considering the very high prevalence in the high risk group.

Introduction

The table seems confusing and unnecessary. This study is meant to describe overall prevalence and genotypes. Not to describe the assessment or management

Methodology

Could benefit from better subtitles.

Results

I found this sections rather confusing with too much data. I appreciated the availability of figures that clarified the study a lot more.

Reviewer #2: This manuscript it's a systematic review of data on prevalence and genotype distribution off hepatitis b virus in Kenya. In all, 25 studies published in English between 2000 and 2021 but included in the analysis. The studies were mostly of low quality. The prevalence studies were divided into 4 groups, namely low risk, moderate risk high risk and very high risk for HBV infection. Information on viral genotype was available in 8 studies.

Though there was enough justification for undertaking this study, the design and data have several shortcomings that limit the applicability of its results. Some of these are listed below

1. For prevalence data, most of the studies were of low quality, with very little attempt at random sampling. Hence, the data carry a risk of being biased. Further, some of the studies were done among patients with liver disease; data from these studies have very little relevance for epidemiology. Two studies used RPHA technique, which is known to be fairly unreliable.

The studies in patients with liver disease (the so called very high risk subgroup) need to be excluded; this will also remove studies that were based on RPHA.

Further, instead of dividing the remaining studies into low-, moderate- and high-risk populations, it may be better to divide these into (i) studies likely to represent general population (the current low risk group), (ii) people living with HIV (the current moderate risk group), (iii) IVDUs and (iv) MSM (the last 2 are currently in high-risk group). This will do away with the artificial risk-based classification, and also account for the results which showed similar positivity rates in the medium and the high risk subgroups.

It may also be useful to consider including only those studies that have quality score above a particular cut-off.

2. The Forest plots show the results of both common effects and random effects models. The authors need to choose one (the random effects model is more appropriate) and show the results of only that.

3. For the data on genotype, the authors start with 9 studies, and then exclude some that included no data (reference 39) and two that had data only for drug resistance mutations (references 46, 47). The above 3 studies should have not even been included in the first place for this part of the analysis. It is unclear from Table 3 or text as to which were the remaining 6 studies that had data on genotype and their sample sizes for this purpose. This needs to be clarified using a separate table for data for this part of the analysis.

4. The paper goes into some other aspects somewhat peripherally, e.g. anti-HBc prevalence, anti-HBs prevalence, HBeAg prevalence, drug resistance mutations, eligibility for treatment, etc. These should not be a part of the paper. Thus, Tables 1 and 7 can be deleted.

5. The paper is too long and can be shortened.

6. PLOS authors have the option to publish the peer review history of their article (what does this mean?). If published, this will include your full peer review and any attached files.

**Do you want your identity to be public for this peer review?** For information about this choice, including consent withdrawal, please see our Privacy Policy.

Reviewer #1: No

Reviewer #2: No

---

## [Decision Letter · Decision Letter 1]

27 Oct 2022

PGPH-D-22-00830R1

A systematic review of Hepatitis B virus (HBV) prevalence and genotypes in Kenya: Data to inform clinical care and health policy

Dear Dr. Matthews,

Thank you for submitting your manuscript to PLOS Global Public Health. After careful consideration, we feel that it has merit but does not fully meet PLOS Global Public Health’s publication criteria as it currently stands. Therefore, we invite you to submit a revised version of the manuscript that addresses the points raised during the review process.

Please see the three additional suggestions from Reviewer 2.  I view the reviewer’s first point about an adjustment to your risk groupings as a suggestion, and not a requirement for publication.

We look forward to receiving your revised manuscript.

Kind regards,

Abraham D. Flaxman, Ph.D.

Academic Editor

Journal Requirements:

Additional Editor Comments (if provided):

Reviewers' comments:

Reviewer's Responses to Questions

**Comments to the Author**

1. If the authors have adequately addressed your comments raised in a previous round of review and you feel that this manuscript is now acceptable for publication, you may indicate that here to bypass the “Comments to the Author” section, enter your conflict of interest statement in the “Confidential to Editor” section, and submit your "Accept" recommendation.

Reviewer #2: All comments have been addressed

2. Does this manuscript meet PLOS Global Public Health’s publication criteria? Is the manuscript technically sound, and do the data support the conclusions? The manuscript must describe methodologically and ethically rigorous research with conclusions that are appropriately drawn based on the data presented.

Reviewer #2: Partly

3. Has the statistical analysis been performed appropriately and rigorously?

Reviewer #2: No

4. Have the authors made all data underlying the findings in their manuscript fully available (please refer to the Data Availability Statement at the start of the manuscript PDF file)?

Reviewer #2: Yes

5. Is the manuscript presented in an intelligible fashion and written in standard English?

Reviewer #2: Yes

6. Review Comments to the Author

Reviewer #2: The revised manuscript is better than the original submission. However, I disagree with the authors on the following points, and still believe that the authors should:

a. The studies should not be divided into groups such as low risk, moderate risk, high risk and very high risk for HBV infection. Even the authors' results show prevalence to be similar in moderate and high risk groups. Therefore, the studies should instead be grouped as: (i) those in groups likely to represent general population (healthy persons, antenatal women, etc -- the current low risk group), (ii) people living with HIV (the current moderate risk group), (iii) IVDUs and (iv)

MSM (the last 2 are currently in high-risk group).

b. Calling studies in 'liver disease patients' as 'high risk' group is wrong. It is not that liver disease is leading to a high risk of liver disease. Instead it is HBV infection which is the cause of liver disease. Hence, data on liver disease patients should not be included in the main table, and should be presented separately.

c. Presenting data on 'studies with genetic data' is irrelevant. What matters it the genotype distribution, as is also clear from the study title ("HBV prevalence and genotypes in Kenya". Hence, the three studies that contain no useful data or data on drug resistance mutations etc should be excluded.

In my opinion, the above changes are mandatory before the paper is accepted for publication. Also, these should help to improve the paper's quality.

7. PLOS authors have the option to publish the peer review history of their article (what does this mean?). If published, this will include your full peer review and any attached files.

**Do you want your identity to be public for this peer review?** For information about this choice, including consent withdrawal, please see our Privacy Policy.

Reviewer #2: No

---

## [Editor Report · Decision Letter 2]

29 Nov 2022

A systematic review of Hepatitis B virus (HBV) prevalence and genotypes in Kenya: Data to inform clinical care and health policy

PGPH-D-22-00830R2

Dear Prof Matthews,

We are pleased to inform you that your manuscript 'A systematic review of Hepatitis B virus (HBV) prevalence and genotypes in Kenya: Data to inform clinical care and health policy' has been provisionally accepted for publication in PLOS Global Public Health.

Best regards,

Abraham D. Flaxman, Ph.D.

Academic Editor